# Hypoxia-inducible factor cell non-autonomously regulates *C. elegans* stress responses and behavior via a nuclear receptor

Corinne L Pender[1,2,3], H Robert Horvitz[1,2,3]*

[1]Department of Biology, Howard Hughes Medical Institute, Massachusetts Institute of Technology, Cambridge, United States; [2]McGovern Institute for Brain Research, Massachusetts Institute of Technology, Cambridge, United States; [3]Koch Institute for Integrative Cancer Research, Massachusetts Institute of Technology, Cambridge, United States

**Abstract** The HIF (hypoxia-inducible factor) transcription factor is the master regulator of the metazoan response to chronic hypoxia. In addition to promoting adaptations to low oxygen, HIF drives cytoprotective mechanisms in response to stresses and modulates neural circuit function. How most HIF targets act in the control of the diverse aspects of HIF-regulated biology remains unknown. We discovered that a HIF target, the *C. elegans* gene *cyp-36A1*, is required for numerous HIF-dependent processes, including modulation of gene expression, stress resistance, and behavior. *cyp-36A1* encodes a cytochrome P450 enzyme that we show controls expression of more than a third of HIF-induced genes. CYP-36A1 acts cell non-autonomously by regulating the activity of the nuclear hormone receptor NHR-46, suggesting that CYP-36A1 functions as a biosynthetic enzyme for a hormone ligand of this receptor. We propose that regulation of HIF effectors through activation of cytochrome P450 enzyme/nuclear receptor signaling pathways could similarly occur in humans.
DOI: https://doi.org/10.7554/eLife.36828.001

*For correspondence:
horvitz@mit.edu

Competing interests: The authors declare that no competing interests exist.

## Introduction

The capacity to sense and respond to oxygen deprivation, or hypoxia, is crucial to normal physiological function and survival of aerobic organisms, which require oxygen to perform respiration and generate energy in the form of ATP. The fundamental importance of a mechanism to detect and react to low oxygen is reflected in the presence of a conserved hypoxia-response pathway in most animal cells. This pathway consists of the transcription factor HIF, or hypoxia-inducible factor, and its negative regulator, the prolyl hydroxylase EGLN, which together mediate a diversity of metabolic and physiological adaptations to hypoxia. The three human EGLNs, which were identified as homologs of the *C. elegans* protein EGL-9, function as oxygen sensors. In the presence of oxygen, EGLN hydroxylates the HIF α-subunit (HIFα), allowing the von Hippel-Lindau (VHL) E3 ubiquitin ligase to promote HIFα degradation (*Maxwell et al., 1999*; *Jaakkola et al., 2001*; *Epstein et al., 2001*; *Ivan et al., 2001*). In conditions of low oxygen, HIFα is stabilized and acts with its partner HIFβ to drive adaptations to hypoxia through activation of its transcriptional targets (*Kaelin and Ratcliffe, 2008*; *Semenza, 2011*; *Wang et al., 1995*).

The canonical function of the EGLN/HIF pathway is to regulate genes that either increase oxygen availability, for example by promoting erythropoiesis and angiogenesis, or reduce the cellular requirement for oxygen, for example by driving a shift from oxidative phosphorylation to glycolytic

metabolism. However, a growing body of work has found roles for the EGLN/HIF pathway in controlling other aspects of animal physiology and behavior. HIF promotes the response to numerous stressors, including infection, proteotoxicity, and oxidative stress (*Palazon et al., 2014*; *Schito and Rey, 2018*; *Nakazawa et al., 2016*; *Powell-Coffman, 2010*). HIF activation is associated with increased lifespan in *C. elegans* (*Mehta et al., 2009*; *Zhang et al., 2009*; *Chen et al., 2009*; *Lee et al., 2010*); this longevity phenotype likely stems from the improved stress resistance associated with HIF activity, as is often the case for pathways regulating longevity (*Leiser et al., 2015*; *Shore and Ruvkun, 2013*). The EGLN/HIF pathway also modulates several behaviors of *C. elegans* following prolonged hypoxia exposure, suggesting a role for this pathway in tuning neural circuit function (*Chang and Bargmann, 2008*; *Pocock and Hobert, 2010*; *Ma et al., 2012*). The mechanisms by which HIF mediates these non-canonical physiological and behavioral changes remain poorly defined.

Here we report the discovery of an endocrine signaling pathway that regulates multiple aspects of physiology and behavior downstream of HIF in *C. elegans*. From a genetic screen for suppressors of an *egl-9(lf)* mutant behavioral defect, we identified a cytochrome P450 gene, *cyp-36A1*, that is required for modulation of egg-laying behavior by the *egl-9/hif-1* pathway. *cyp-36A1* is transcriptionally upregulated in hypoxia or *egl-9(lf)* mutants, in which HIF-1 is constitutively active, and appears to be a direct HIF-1 target. *cyp-36A1* controls expression of more than a third of HIF-1-upregulated genes, demonstrating that *cyp-36A1* acts broadly downstream of *hif-1*. Regulation of gene expression and behavior by *cyp-36A1* occurs cell non-autonomously, and the downstream effector of *cyp-36A1* is the nuclear hormone receptor *nhr-46*, indicating that the likely function of CYP-36A1 is to generate a diffusible signal that controls NHR-46 activity. In addition to modulating behavior and gene expression, *cyp-36A1* and *nhr-46* mediate multiple forms of stress resistance associated with HIF activation. We conclude that CYP-36A1 and NHR-46 are important downstream effectors of the EGL-9/HIF pathway and function together to regulate a wide range of HIF-mediated physiology.

## Results

### A screen for suppressors of the *egl-9(lf)* egg-laying defect identifies the cytochrome P450 gene *cyp-36A1*

To identify novel, functionally important HIF effectors, we analyzed the modulation of *C. elegans* egg laying, the behavior that led our laboratory to discover the first EGLN gene, *egl-9*, and the first known functional role for any member of the EGLN/HIF pathway (*Trent et al., 1983*). *egl-9(lf)* mutants, in which HIF-1 is constitutively active, are defective in egg laying and become bloated with eggs as adults. Although the egg-laying defect of *egl-9(lf)* mutants is well-established, the downstream effectors of EGL-9 and HIF-1 in regulating egg-laying behavior remain unknown.

We performed a mutagenesis screen to identify genes that act in response to *egl-9* to control egg laying. Specifically, we screened for second-site mutations that suppressed the egg-laying defect of *egl-9(lf)* animals (*Figure 1A*). Such suppressors could define genes that function downstream of *egl-9*. Two isolates from this screen were allelic to *hif-1* (*Figure 1—figure supplement 1A and B*), consistent with a previous observation that *hif-1(lf)* suppresses the *egl-9(lf)* egg-laying defect (*Bishop et al., 2004*) and validating the screen as a means of identifying components of the HIF-1 pathway. A third isolate, *n5666*, was not allelic to *hif-1* and had a G106R missense mutation in the gene *cyp-36A1*, which encodes a cytochrome P450 enzyme (*Figure 1—figure supplement 1A and C*). A transgene carrying a wild-type copy of *cyp-36A1* fully rescued the suppression by *n5666* of the *egl-9(lf)* egg-laying defect, demonstrating that the mutation in *cyp-36A1* is the causative mutation and suggesting that the suppression phenotype is caused by reduction of *cyp-36A1* function (*Figure 1B–1E*). Confirming these conclusions, a nonsense allele of *cyp-36A1* also suppressed the *egl-9(lf)* egg-laying defect (*Figure 3—figure supplement 1B and C*). *cyp-36A1(lf)* single mutants did not exhibit hyperactive egg-laying behavior (*Figure 1F* and *Figure 3—figure supplement 1A and D*), indicating that suppression of the *egl-9(lf)* egg-laying defect by *cyp-36A1(lf)* is not a consequence of a nonspecific increase in egg-laying rate. We further showed that *cyp-36A1(lf)* suppressed the previously reported egg-laying defect of hypoxia-exposed worms (*Miller and Roth, 2009*), demonstrating a role for CYP-36A1 under physiological conditions of HIF-1 activation (*Figure 1—figure supplement 2*). We then analyzed the role of *cyp-36A1* in regulating other behaviors. We observed that *egl-9(lf)* mutants have reduced locomotion and defecation rates, both of which were suppressed

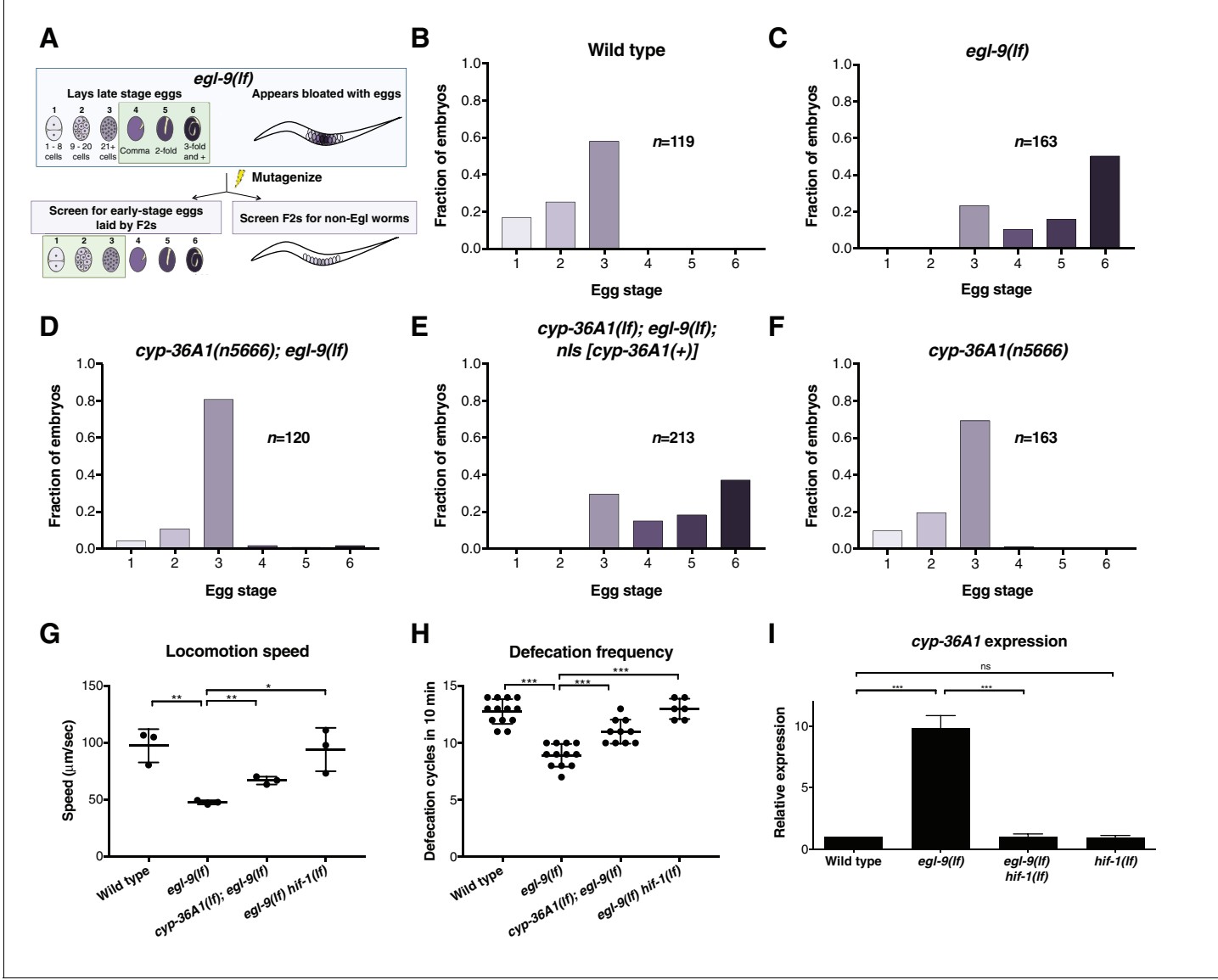

**Figure 1.** The cytochrome P450 gene *cyp-36A1* is an effector of the hypoxia-response pathway that regulates behavior. (A) Schematic of the screen design. Stages of embryonic development adapted from *Ringstad and Horvitz (2008)* and *Paquin et al. (2016)*. (B–F) Distribution of stages of eggs newly laid by adult hermaphrodites, used as a proxy for egg retention time *in utero*, of animals of the indicated genotypes. (B) Stages of eggs laid by wild-type animals. (C) *egl-9* loss-of-function (*lf*) mutants laid later stage eggs than the wild type (p<0.001, Chi-square test with Holm-Bonferroni correction). (D) *cyp-36A1(n5666)* suppressed the egg-laying defect of *egl-9(lf)* mutants (p<0.001). (E) The *cyp-36A1(+)* transgene, which contains wild-type *cyp-36A1*, rescued the suppression of the egg-laying defect observed in *cyp-36A1(n5666); egl-9(lf)* mutants (p<0.001). (F) *cyp-36A1(n5666)* mutants displayed wild-type egg laying (p>0.05). (G) *egl-9(lf)* mutants were defective in locomotion rate, and this defect was suppressed by *hif-1(lf)* and *cyp-36A1(lf)* mutations. Mean ± SD of *n* = 3 biological replicates, *p<0.05, **p<0.01 considered significant (Student's t-test with Holm-Bonferroni correction). (H) *egl-9(lf)* mutants were defective in defecation rate, and this defect was suppressed by *hif-1(lf)* and *cyp-36A1(lf)*. Mean ± SD of *n* ≥ 6 animals, ***p<0.001 considered significant (Student's t-test with Holm-Bonferroni correction). (I) Relative expression of *cyp-36A1* mRNA in the wild type, *egl-9(lf)*, *egl-9(lf) hif-1(lf)*, and *hif-1(lf)* mutants, measured by qRT-PCR and normalized to the expression of the large ribosomal subunit *rpl-32*. Mean ± SD of *n* = 3 biological replicates, ***p<0.001 considered significant. ns, not significant (Student's t-test with Holm-Bonferroni correction). Alleles used for (B–F) were *cyp-36A1(n5666)*, *egl-9(n586)*, and *nIs674* (*nIs* [*cyp-36A1(+)*]), and all strains used in (B–F) contained the *nIs470* (*P_cysl-2_::gfp*) transgene. Alleles used for (G–I) were *egl-9(sa307)*, *hif-1(ia4)*, and *cyp-36A1(gk824636)*.

DOI: https://doi.org/10.7554/eLife.36828.002

The following figure supplements are available for figure 1:

**Figure supplement 1.** A screen for suppressors of the *egl-9(lf)* egg-laying defect.

DOI: https://doi.org/10.7554/eLife.36828.003

**Figure supplement 2.** *cyp-36A1(lf)* suppresses the egg-laying defect of hypoxia-exposed animals.

*Figure 1 continued on next page*

*Figure 1 continued*

DOI: https://doi.org/10.7554/eLife.36828.004

by *hif-1(lf)* (*Figure 1G and H*). *cyp-36A1(lf)* partially suppressed the slow locomotion and defecation rates of *egl-9(lf)* mutants, showing that CYP-36A1 modulates not only egg laying but also multiple other HIF-1-regulated behaviors.

Next we observed that *cyp-36A1* expression is increased in *egl-9(lf)* mutants in a *hif-1*-dependent manner (*Figure 1I*), consistent with results from an earlier genome-wide microarray study that identified *cyp-36A1* as one of 63 genes regulated by *hif-1* in hypoxia-exposed worms (*Shen et al., 2005*). ChIP-seq of HIF-1 by the modERN project showed HIF-1 binding at two sites near the *cyp-36A1* coding region, one 5' to the start of the gene and one in the first intron (*Kudron et al., 2018*); both of these sites contain the HIF binding motif 5'RCGTG (*Kaelin and Ratcliffe, 2008*). We conclude that *cyp-36A1* is a downstream effector of the hypoxia-response pathway that regulates multiple behaviors and that *cyp-36A1* likely is a direct transcriptional target of HIF-1.

## CYP-36A1 regulates gene expression changes and stress resistance downstream of HIF-1

We sought to determine if CYP-36A1 regulates other HIF-1-dependent processes. Based on sequence identity, CYP-36A1 is most closely related to the CYP2 family of cytochrome P450 enzymes, which function in both detoxification of xenobiotics and metabolism of endogenous molecules (*Nebert et al., 2013*). CYP2 family members and other CYPs can act on endogenous substrates to generate diffusible signaling molecules that regulate gene expression, such as eicosanoids and steroid hormones (*Rendic and Guengerich, 2015*; *Dennis and Norris, 2015*; *Evans and Mangelsdorf, 2014*). We hypothesized that CYP-36A1 might function in a transcriptional cascade to mediate aspects of HIF-1-dependent gene regulation. We performed an RNA-seq experiment comparing the wild type, *egl-9(lf)* mutants, *egl-9(lf) hif-1(lf)* double mutants, and *cyp-36A1(lf); egl-9(lf)* double mutants. We found that *hif-1(lf)* suppressed *egl-9*-dependent gene expression for 93% of *egl-9(lf)*-downregulated genes and 87% of *egl-9(lf)*-upregulated genes, indicating that most but not all regulation of transcription by *egl-9* occurs through *hif-1*, consistent with previous work (*Angeles-Albores et al., 2018*). We further found that 36% of HIF-1-upregulated genes (i.e. genes that are upregulated in *egl-9(lf)* mutants and suppressed by *hif-1(lf)*) and 10% of HIF-1-downregulated genes were also regulated by *cyp-36A1* (*Figure 2A and B* and *Supplementary files 1* and *2*). We focused on the HIF-1-upregulated genes, for which CYP-36A1 function appeared to be more broadly required. Gene ontology (GO) enrichment analysis of these HIF-1/CYP-36A1-upregulated genes suggested a role for *cyp-36A1* in regulating stress resistance downstream of *egl-9* and *hif-1* (*Figure 2—source data 1*). The EGL-9/HIF-1 pathway has previously been implicated in responses to numerous stressors in both nematodes and mammals, with crosstalk occurring between HIF and regulators of the immune response, unfolded protein response, and other stress-response pathways (*Palazon et al., 2014*; *Schito and Rey, 2018*; *Wouters and Koritzinsky, 2008*; *Nakazawa et al., 2016*; *Powell-Coffman, 2010*). We tested whether CYP-36A1 is involved in the response to three stressors for which HIF-1 is known to mediate resistance in *C. elegans*: infection by the pathogenic bacteria *Pseudomonas aeruginosa* (*Darby et al., 1999*; *Bellier et al., 2009*; *Shao et al., 2010*; *Budde and Roth, 2011*; *Kirienko et al., 2013*), tunicamycin-induced ER stress (*Leiser et al., 2015*), and oxidative stress from tert-butyl hydroperoxide (*Bellier et al., 2009*). Animals in which HIF-1 is constitutively active because of mutation in *egl-9* or the *C. elegans* VHL homolog *vhl-1* are resistant to these stressors relative to wild-type animals: such mutants survive longer when grown on *Pseudomonas aeruginosa* strain PA14 (*Bellier et al., 2009*), display reduced tunicamycin-induced growth inhibition (*Leiser et al., 2015*), and survive exposure to tert-butyl hydroperoxide at a higher rate than the wild type (*Bellier et al., 2009*). We found that *cyp-36A1(lf); egl-9(lf)* double mutants are more sensitive than *egl-9(lf)* mutants to all three of these stressors (*Figure 2C–2E* and *Figure 2—figure supplement 1*), suggesting that CYP-36A1 mediates responses to these stressors downstream of HIF-1. Together the CYP-36A1-dependent changes in gene expression and stress resistance indicate that CYP-36A1 plays a major role in regulating HIF-1-mediated physiology.

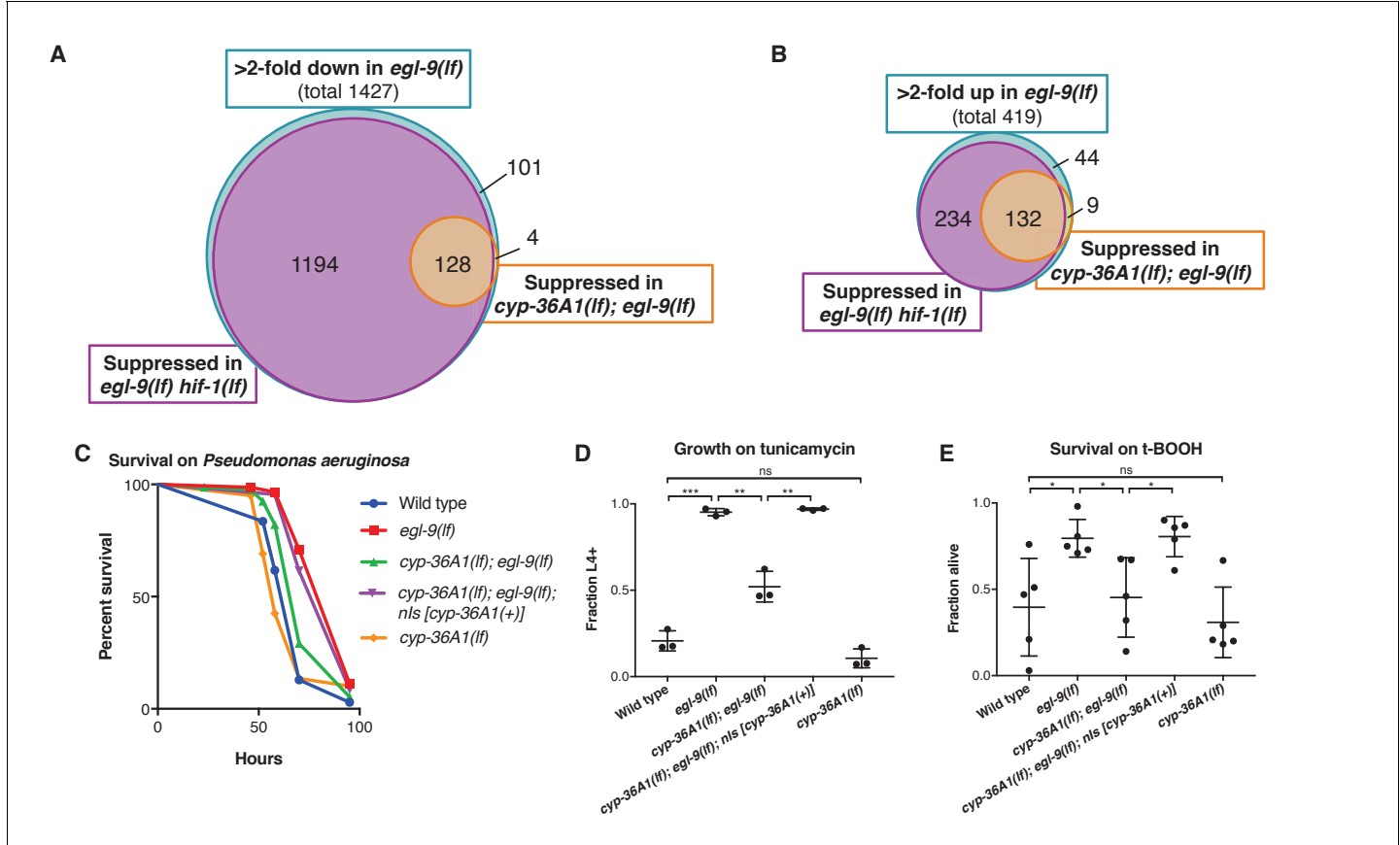

**Figure 2.** CYP-36A1 acts downstream of HIF-1 to regulate gene expression changes and stress responses. (**A**) Blue circle: Genes that were at least twofold downregulated in *egl-9(lf)* mutants. Purple and orange circles: Subset of *egl-9(lf)*-downregulated genes that were significantly upregulated in *egl-9(lf) hif-1(lf)* double mutants (purple) or *cyp-36A1(lf); egl-9(lf)* double mutants (orange) vs. *egl-9(lf)* single mutants. (**B**) Blue circle: Genes that were at least twofold upregulated in *egl-9(lf)* mutants. Purple and orange circles: Subset of *egl-9(lf)*-upregulated genes that were significantly downregulated in *egl-9(lf) hif-1(lf)* double mutants (purple) or *cyp-36A1(lf); egl-9(lf)* double mutants (orange) vs. *egl-9(lf)* single mutants. Significance for all comparisons in (**A**) and (**B**) was based on two biological replicates and determined by the Benjamini-Hochberg procedure with a false-discovery rate of 0.05. (**C**) Survival of animals grown from the L4 larval stage on the pathogen *Pseudomonas aeruginosa*. Wild type (*n* = 86 animals) vs. *egl-9(lf)* (*n* = 154) p<0.001; *egl-9(lf)* vs. *cyp-36A1(lf); egl-9(lf)* (*n* = 83), p<0.001; *cyp-36A1(lf); egl-9(lf)* vs. *cyp-36A1(lf); egl-9(lf); nIs [cyp-36A1(+)]* (*n* = 97), p<0.001; wild type vs. *cyp-36A1(lf)* (*n* = 61) p<0.05, as determined by the log-rank (Mantel-Cox) test, correcting for multiple comparisons with the Holm-Bonferroni method. See figure supplement for replicate data. (**D**) Survival of animals to the L4 larval stage or later after growth for three days from the L1 larval stage on plates containing 5 µg/ml tunicamycin. Mean ± SD of *n* = 3 biological replicates. **p<0.01, ***p<0.001 considered significant. ns (p>0.05), not significant (Student's t-test with Holm-Bonferroni correction). (**E**) Survival of animals exposed to 7.5 mM tert-butyl hydroperoxide for 10 hr as young adults. Mean ± SD of *n* = 5 biological replicates. *p<0.05 considered significant. ns (p>0.05), not significant (Student's t-test with Holm-Bonferroni correction). Alleles used for (**A**) and (**B**) were *egl-9(sa307)*, *hif-1(ia4)*, and *cyp-36A1(gk824636)*. Alleles used for (**C–E**) were *egl-9(n586)*, *cyp-36A1(n5666)*, and *nIs674* (*nIs [cyp-36A1(+)]*), and all strains used in (**C–E**) contained the *nIs470 (P_{cysl-2}::gfp)* transgene.

DOI: https://doi.org/10.7554/eLife.36828.005

The following source data and figure supplement are available for figure 2:

**Source data 1.** RNA-seq GO enrichment analysis.
DOI: https://doi.org/10.7554/eLife.36828.007
**Figure supplement 1.** Replicate data for survival on *Pseudomonas aeruginosa*.
DOI: https://doi.org/10.7554/eLife.36828.006

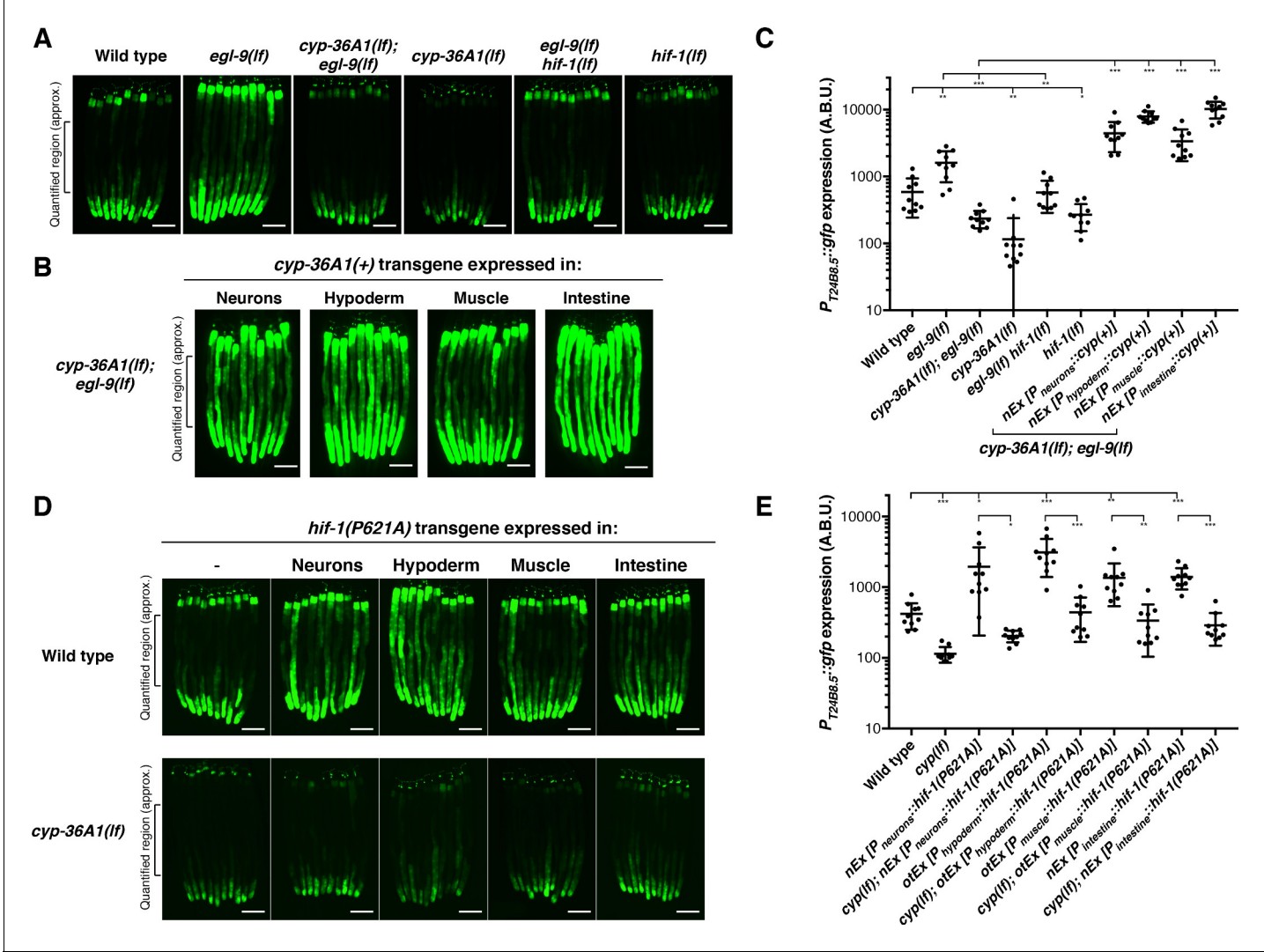

**Figure 3.** HIF-1 and CYP-36A1 cell non-autonomously regulate expression of a stress-responsive gene. (**A**) $P_{T24B8.5}$::*gfp* expression of the indicated genotypes (*n* = 10 animals per image). Scale bars, 100 μm. (**B**) Expression of *cyp-36A1(+)* specifically in neurons, hypoderm, muscle, or intestine of *cyp-36A1(lf); egl-9(lf)* animals increased expression of $P_{T24B8.5}$::*gfp* in intestine (*n* = 10 animals per image). Scale bars, 100 μm. (**C**) Quantification of fluorescence intensity for (**A**) and (**B**), measured as average intensity for a 300 μm section of the intestine in the midbody of each animal, as indicated. *p<0.05, **p<0.01, ***p<0.001 considered significant (Student's t-test with Holm-Bonferroni correction). Mean ± SD of *n* = 10 animals. See figure supplement for replicate data. (**D**) Expression of *hif-1(P621A)*, which encodes a stable variant of HIF-1 (*Pocock and Hobert, 2008*), specifically in neurons, hypoderm, muscle, or intestine increased expression of $P_{T24B8.5}$::*gfp* in intestine; increased expression was suppressed by *cyp-36A1(lf)* (*n* = 10 animals per image). Scale bars, 100 μm. (**E**) Quantification of fluorescence intensity for (**D**), measured as average intensity for a 300 μm section of the intestine in the midbody of each animal, as indicated. *p<0.05, ***p<0.01, ***p<0.001 considered significant (Student's t-test with Holm-Bonferroni correction). Mean ± SD of *n* = 10 animals. See figure supplement for replicate data. Alleles used were *egl-9(sa307)*, *hif-1(ia4)*, *cyp-36A1(gk824636)*, *nEx2699[P_neurons::hif-1(P621A)]*, *otEx3156 [P_hypoderm::hif-1(P621A)]*, *otEx3165 [P_muscle::hif-1(P621A)]*, *nEx2860 [P_intestine:: hif-1(P621A)]*, *nEx2853 [P_neurons::cyp (+)]*, *nEx2856 [P_hypoderm::cyp(+)]*, *nEx2859 [P_muscle::cyp(+)]*, and *nEx2849 [P_intestine::cyp(+)]*. All strains contained the *agIs219* ($P_{T24B8.5}$::*gfp*) transgene.
DOI: https://doi.org/10.7554/eLife.36828.008

The following figure supplements are available for figure 3:

**Figure supplement 1.** *cyp-36A1* expression in multiple tissues rescues the egg-laying phenotype of *cyp-36A1(lf); egl-9(lf)*.
DOI: https://doi.org/10.7554/eLife.36828.009

**Figure supplement 2.** *cyp-36A1* is expressed in many tissues.
DOI: https://doi.org/10.7554/eLife.36828.010

**Figure supplement 3.** Replicate data for $P_{T24B8.5}$::*gfp* reporter expression.
DOI: https://doi.org/10.7554/eLife.36828.011

**Figure supplement 4.** Replicate data for $P_{T24B8.5}$::*gfp* reporter expression.

*Figure 3 continued on next page*

*Figure 3 continued*

DOI: https://doi.org/10.7554/eLife.36828.012

## CYP-36A1 functions cell non-autonomously to regulate gene expression

We next sought to identify the site of action of CYP-36A1. We hypothesized that CYP-36A1 might function cell non-autonomously, as is the case for other cytochrome P450 enzymes that generate signaling molecules (*Nebert et al., 2013*; *Evans and Mangelsdorf, 2014*; *Gerisch and Antebi, 2004*). We observed *cyp-36A1* expression in many tissues, including neurons, intestine, hypoderm, and muscle (*Figure 3—figure supplement 2*). To test the hypothesis of cell non-autonomous CYP-36A1 function, we focused on a *cyp-36A1*-mediated abnormality of *egl-9(lf)* mutants for which the site of dysfunction is well defined. Specifically, we examined expression of a GFP transcriptional reporter for the gene *T24B8.5* (*Shivers et al., 2009*), which is expressed in only the intestine and based on our RNA-seq data is upregulated in *egl-9(lf)* mutants in a *cyp-36A1*-dependent manner. Interestingly, *T24B8.5* expression is also upregulated in response to infection, ER stress, and oxidative stress (*Shivers et al., 2009*; *Lim et al., 2014*; *Park et al., 2009*). Expression of the reporter recapitulated the *T24B8.5* expression changes observed by RNA-seq: increased expression of GFP was observed in *egl-9(lf)* mutants, which was suppressed by a second mutation in either *hif-1* or *cyp-36A1* (*Figure 3A and C* and *Figure 3—figure supplement 3*). To determine the site of action of *cyp-36A1* for regulation of intestinal *T24B8.5* expression, we expressed wild-type *cyp-36A1* cDNA using tissue-specific promoters. We found that the low $P_{T24B8.5}::gfp$ expression of *cyp-36A1(lf); egl-9(lf)* double mutants was rescued by expressing *cyp-36A1(+)* either cell autonomously in the intestine or cell non-autonomously in neurons, hypoderm or body-wall muscle (*Figure 3B and C* and *Figure 3—figure supplement 3*). *cyp-36A1(+)* expression in all four tissues also rescued the suppression of the egg-laying defect of *cyp-36A1(lf); egl-9(lf)* mutants (*Figure 3—figure supplement 1*). Next we found that expressing a nondegradable constitutively active HIF-1 mutant protein (P621A) (*Pocock and Hobert, 2008*) in any of the same four tissues also promoted intestinal expression of the GFP reporter and that this HIF-1-mediated increase in expression required *cyp-36A1* (*Figure 3D and E* and *Figure 3—figure supplement 4*). Thus, CYP-36A1 can function cell non-autonomously to regulate gene expression downstream of HIF-1, consistent with the hypothesis that CYP-36A1 acts by generating a diffusible signal.

## A screen for suppressors of *cyp-36A1(lf)* identifies the nuclear receptor gene *nhr-46*

We performed a mutagenesis screen to identify CYP-36A1 effectors that regulate egg-laying behavior, stress responses, and gene expression. We screened for mutations that suppressed both the low $P_{T24B8.5}::gfp$ expression and normal egg laying of *cyp-36A1(lf); egl-9(lf)* double mutants, looking for triple mutants that, like *egl-9(lf)* single mutants, had high GFP expression and were egg-laying defective. By screening for suppressors of the two abnormalities simultaneously, we were able to focus on effectors of CYP-36A1 rather than finding genes that affect only egg laying or only expression of *T24B8.5* independently of the EGL-9/HIF-1/CYP-36A1 pathway. From this screen we identified one nonsense and one missense allele of the nuclear receptor gene *nhr-46* (*Figure 4A–4D* and *Figure 4—figure supplement 1*), both of which caused an egg-laying defect and high expression of the $P_{T24B8.5}::gfp$ reporter. We tested whether *nhr-46* also functions in regulating stress responses downstream of *cyp-36A1* and found that *cyp-36A1(lf); nhr-46(lf); egl-9(lf)* triple mutants were more resistant to *Pseudomonas* infection, ER stress, and oxidative stress than *cyp-36A1(lf); egl-9(lf)* double mutants (*Figure 4E–4G* and *Figure 4—figure supplement 2*). Thus, NHR-46 is a downstream effector of CYP-36A1 in regulation of stress resistance as well as of behavior and gene expression. Interestingly, *nhr-46(lf)* single mutants displayed wild-type egg laying (*Figure 4D*), tunicamycin resistance (*Figure 4F*), and oxidative stress resistance (*Figure 4G*), and nearly wild-type survival on *Pseudomonas* (*Figure 4E* and *Figure 4—figure supplement 2*), indicating that in addition to *nhr-46* at least one other pathway is required to transduce *egl-9*-mediated modulation of egg laying and stress resistance. For example, in *egl-9(lf)* mutants or hypoxia, HIF-1 might drive expression of two (or more) targets that act together to inhibit egg laying and promote stress resistance. Thus, *cyp-36A1* activation and consequent *nhr-46* inhibition would promote egg-laying inhibition and stress

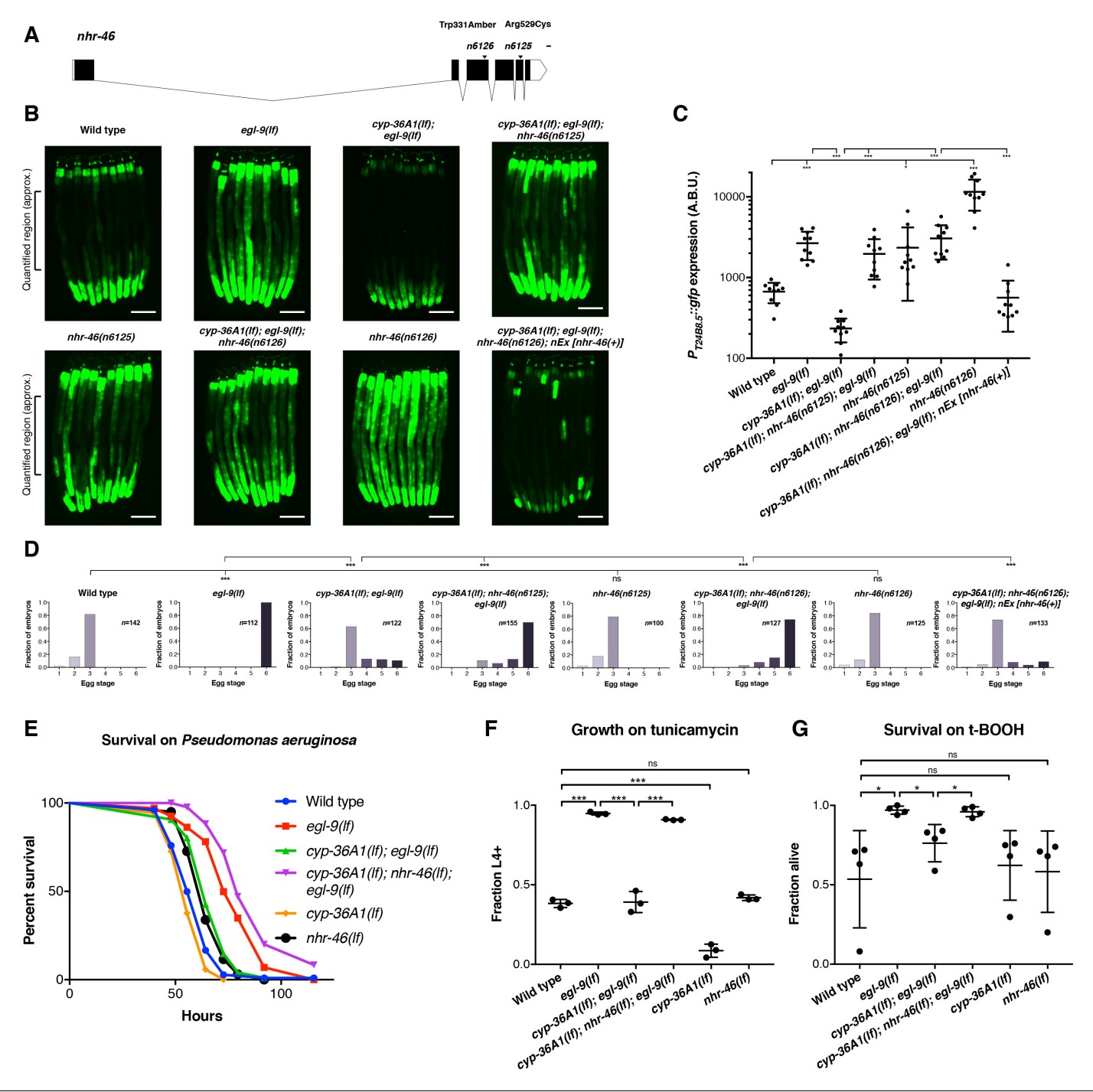

**Figure 4.** The nuclear hormone receptor NHR-46 acts downstream of CYP-36A1. (**A**) *nhr-46* gene diagram; isoform *C45E5.6b* is shown. *n6125* and *n6126* are in the NHR-46 ligand-binding domain. Scale bar, 100 bases. (**B**) $P_{T24B8.5}$::*gfp* fluorescence of the indicated genotypes (*n* = 10 animals per image). Scale bars, 100 µm. (**C**) Quantification of fluorescence intensity for (**B**), measured as average intensity for a 300 µm section of the intestine in the midbody of each animal, as indicated. *p<0.05, ***p<0.001 considered significant (Student's t-test with Holm-Bonferroni correction). Mean ±SD of *n* = 10 animals. See figure supplement for replicate data. (**D**) Distribution of stages of eggs newly laid by adult hermaphrodites of the indicated genotypes. ***p<0.001 considered significant. ns (p>0.05), not significant (Chi-square test with Holm-Bonferroni correction). (**E**) Survival of animals grown from the L4 larval stage on the pathogen *Pseudomonas aeruginosa*. Wild type (*n* = 129) vs. *egl-9(lf)* (*n* = 67) p<0.001; *egl-9(lf)* vs. *cyp-36A1(lf); egl-9(lf)* (*n* = 129), p<0.001; *cyp-36A1(lf); egl-9(lf)* vs. *cyp-36A1(lf); nhr-46(lf); egl-9(lf)* (n = 94), p<0.001; wild type vs. *cyp-36A1(lf)* (*n* = 106), p<0.05; wild type vs. *nhr-46(lf)* (*n* = 121), p<0.001, as determined by the log-rank (Mantel-Cox) test, correcting for multiple comparisons with the Holm-Bonferroni method. *egl-9(lf)* allele was *egl-9(n586)*. See figure supplement for replicate data. (**F**) Survival of animals to the L4 larval stage or later after growth for

*Figure 4 continued on next page*

*Figure 4 continued*

three days from the L1 larval stage on plates containing 5 μg/ml tunicamycin. Mean ± SD of *n* = 3 biological replicates. \*\*\*p<0.001 considered significant. ns (p>0.05), not significant (Student's t-test with Holm-Bonferroni correction). (G) Survival of animals exposed to 7.5 mM tert-butyl hydroperoxide for 10 hr as young adults. Mean ± SD of *n* = 4 biological replicates. \*p<0.05 considered significant. ns, not significant (Student's t-test with Holm-Bonferroni correction). Alleles used for (B–G) were *egl-9(sa307)*, *cyp-36A1(gk824636)*, *nhr-46(n6126)*, and *nEx2586 (nEx [nhr-46(+)])* except where otherwise noted. All strains in (B), (C), (D), (F), and (G) contained the *agIs219* (P$_{T24B8.5}$::*gfp*) transgene.
DOI: https://doi.org/10.7554/eLife.36828.013

The following figure supplements are available for figure 4:

**Figure supplement 1.** Replicate data for P$_{T24B8.5}$::*gfp* reporter expression.
DOI: https://doi.org/10.7554/eLife.36828.014
**Figure supplement 2.** Replicate data for survival on *Pseudomonas aeruginosa*.
DOI: https://doi.org/10.7554/eLife.36828.015

resistance in *egl-9(lf)* mutant animals – in which this second pathway was also activated – but not in *nhr-46(lf)* single mutants – in which this second pathway was not activated. *nhr-46(lf)* single mutants had increased expression of the P$_{T24B8.5}$::*gfp* reporter (*Figure 4B and C* and *Figure 4—figure supplement 1*), suggesting that at least some gene expression changes can be mediated by *nhr-46* alone.

## *nhr-46* functions tissue-specifically to regulate gene expression and behavior

*nhr-46* is expressed in many tissues, including neurons, hypoderm, muscle, intestine, and the spermatheca (*Feng et al., 2012*). Tissue-specific expression of *nhr-46* in the intestine, but not in neurons or muscle, rescued the high GFP expression caused by *nhr-46(lf)*, indicating that *nhr-46* acts cell autonomously in the intestine to control intestinal expression of *T24B8.5* (*Figure 5A and B* and *Figure 5—figure supplement 1*). *nhr-46* expression in neurons fully rescued the egg-laying defect of *cyp-36A1(lf); nhr-46(lf); egl-9(lf)* triple mutants (*Figure 5C*), demonstrating that *nhr-46* function in neurons is sufficient to regulate egg-laying behavior. *nhr-46* expression in intestine, but not muscle, also partially rescued the egg-laying defect of *cyp-36A1(lf); nhr-46(lf); egl-9(lf)* triple mutants (*Figure 5C*). In combination with the tissue-specific CYP-36A1 and HIF-1 experiments described above, these results suggest that a CYP-36A1-regulated cell non-autonomous signal from any tissue can act on NHR-46 in the intestine to drive intestinal *T24B8.5* expression and in either the nervous system or the intestine to regulate egg-laying behavior. That *nhr-46* can act in either of two different tissues to regulate egg laying suggests that a second intercellular signal, for example a peptide or other hormone, might regulate egg laying downstream of NHR-46.

## Discussion

These studies define a novel molecular genetic pathway that mediates cell non-autonomous regulation of gene expression by the HIF-1 transcription factor. Our genetic analysis indicates that *hif-1* activates the cytochrome P450 *cyp-36A1*, which in turn inhibits the nuclear receptor *nhr-46* (*Figure 6A*). We speculate that the molecular function of CYP-36A1 is to generate an unidentified hormone that binds and regulates NHR-46, similar to other cytochrome P450 enzymes that function upstream of nuclear receptors (*Evans and Mangelsdorf, 2014*) and consistent with our observed cell non-autonomous function of CYP-36A1. We propose the following model (*Figure 6B*): In wild-type animals, EGL-9 inhibits HIF-1 activity, such that the HIF-1 target *cyp-36A1* is not transcribed. The unliganded NHR-46 represses expression of genes that promote stress resistance and inhibit egg laying. In *egl-9(lf)* mutants, as in hypoxia-exposed worms, HIF-1 is stabilized and drives increased *cyp-36A1* expression. A CYP-36A1-generated hormone then binds NHR-46 and antagonizes the repressive function of NHR-46, accounting for the observed negative regulatory relationship between *cyp-36A1* and *nhr-46*. Ligand-bound NHR-46 is likely activated to promote the expression of target genes, by analogy to a well-established mechanism of nuclear receptor regulation in which ligand binding mediates a switch from repressive to activating nuclear receptor function (*Evans and Mangelsdorf, 2014*). Alternatively, CYP-36A1 might degrade a ligand that activates NHR-46.

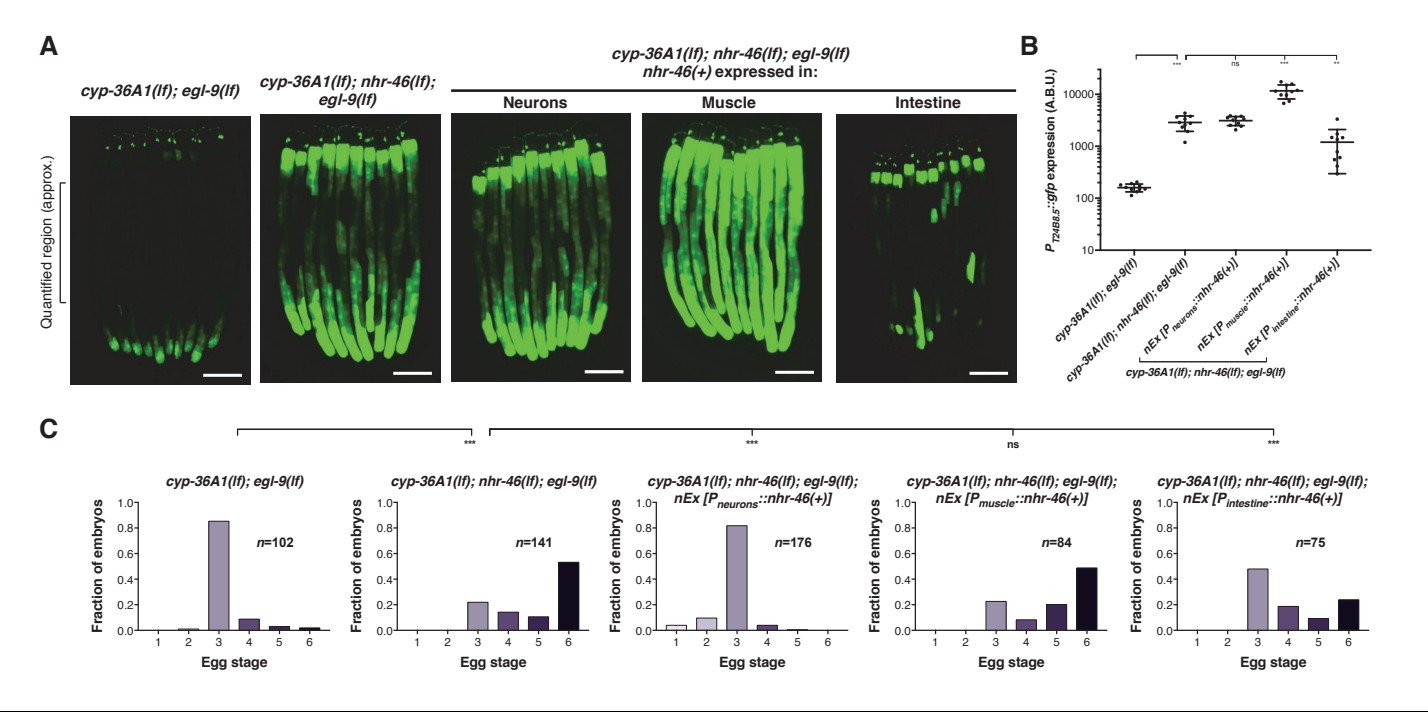

**Figure 5.** NHR-46 acts in different tissues to regulate *T24B8.5* expression and egg laying. (**A**) Expression of *nhr-46(+)* in the intestine but not in neurons or muscle rescued the high $P_{T24B8.5}$::*gfp* expression in the intestine of *cyp-36A1(lf); nhr-46(lf); egl-9(lf)* mutants (n = 10 animals per image). Scale bars, 100 μm. (**B**) Quantification of fluorescence intensity for (**A**), measured as average intensity for a 300 μm section of the intestine in the midbody of each animal, as indicated. \*\*p<0.01, \*\*\*p<0.001 considered significant. ns (p>0.05), not significant (Student's t-test with Holm-Bonferroni correction). Mean ±SD of n = 10 animals. See figure supplement for replicate data. (**C**) Distribution of stages of eggs laid by adult hermaphrodites. Expression of *nhr-46(+)* in the intestine partially rescued the egg-laying defect of *cyp-36A1(lf); nhr-46(lf); egl-9(lf)* mutants. Neuronal *nhr-46(+)* expression fully rescued the egg-laying defect. Expression of *nhr-46(+)* in muscle did not rescue the egg-laying defect of *cyp-36A1(lf); nhr-46(lf); egl-9(lf)* mutants. \*\*\*p<0.001 considered significant. ns (p>0.05), not significant (Chi-square test with Holm-Bonferroni correction). Alleles used were *egl-9(sa307)*, *cyp-36A1 (gk824636)*, *nhr-46(n6126)*, *nEx2713 [$P_{neurons}$::nhr-46(+)]*, *nEx2715 [$P_{muscle}$::nhr-46(+)]*, and *nEx2864 [$P_{intestine}$::nhr-46(+)]*. All strains contained the *agIs219* ($P_{T24B8.5}$::*gfp*) transgene.

DOI: https://doi.org/10.7554/eLife.36828.016

The following figure supplement is available for figure 5:

**Figure supplement 1.** Replicate data for $P_{T24B8.5}$::*gfp* reporter expression.
DOI: https://doi.org/10.7554/eLife.36828.017

## Cell non-autonomous regulation of stress resistance by HIF involves multiple pathways

Numerous studies have reported that HIF-1 promotes longevity and stress resistance of *C. elegans* (*Darby et al., 1999*; *Treinin et al., 2003*; *Bellier et al., 2009*; *Mehta et al., 2009*; *Zhang et al., 2009*; *Chen et al., 2009*; *Lee et al., 2010*; *Shao et al., 2010*; *Budde and Roth, 2011*; *Kirienko et al., 2013*; *Fawcett et al., 2015*). Nonetheless, despite substantial interest in the role of this pathway in stress biology, few relevant HIF effectors have been identified. Interestingly, a recent study reported that HIF-dependent serotonin signaling from the nervous system cell non-autonomously drives expression of the xenobiotic detoxification enzyme flavin-containing monooxygenase-2 (FMO-2) in the intestine, resulting in increased stress resistance and consequent extension of lifespan (*Leiser et al., 2015*). Here we report that a different signal, likely a lipophilic hormone, acts cell non-autonomously downstream of HIF to regulate gene expression and stress resistance. Consistent with the findings of *Leiser et al. (2015)*, our RNA-seq data showed that in *egl-9(lf)* mutants there is a strong induction of *fmo-2* expression and that this induction is suppressed by *hif-1* mutation (*Supplementary file 2*), that is HIF-1 upregulates *fmo-2* expression. Notably, this HIF-dependent expression of *fmo-2* does not require *cyp-36A1*: the high *fmo-2* expression of *egl-9(lf)* mutants is not suppressed by *cyp-36A1(lf)*. We therefore suggest that serotonin-mediated *fmo-2* expression and

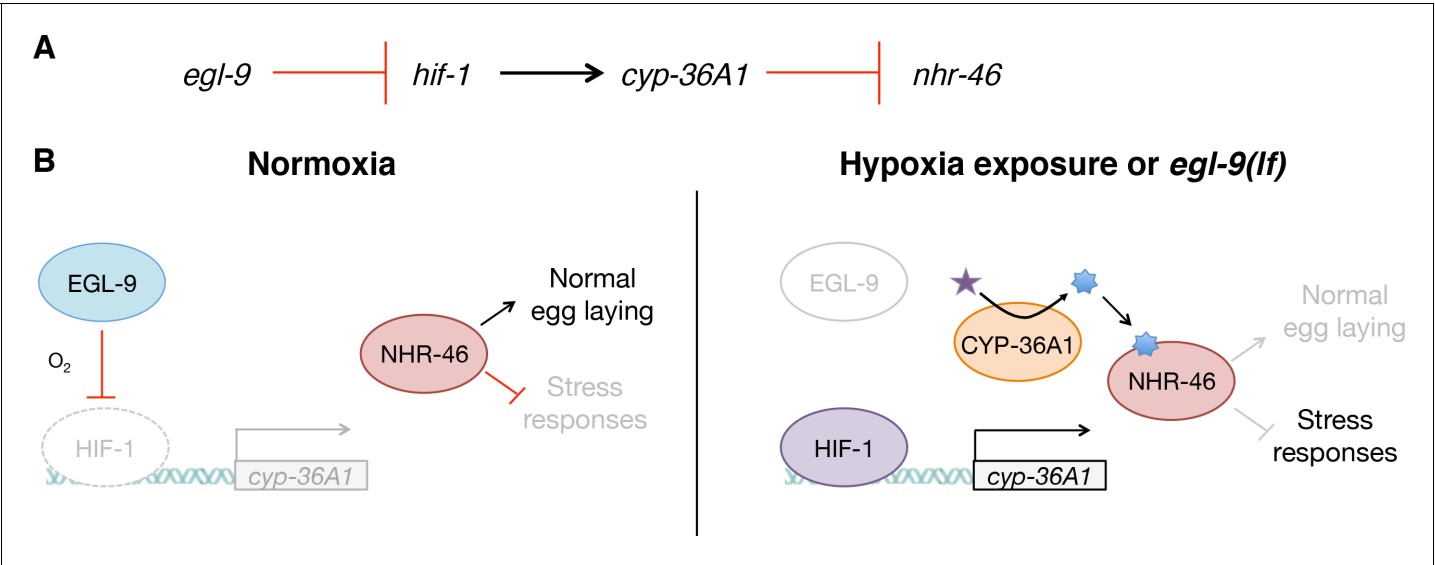

**Figure 6.** Model for the regulation of physiology and behavior by NHR-46 and CYP-36A1. (**A**) The genetic pathway in which *egl-9* inhibits *hif-1*, which activates *cyp-36A1*, which in turn inhibits *nhr-46*. (**B**) Model for how CYP-36A1 and NHR-46 function downstream of HIF-1. We suggest that CYP-36A1, which is transcriptionally upregulated by HIF-1, generates a hormone that binds NHR-46, thereby promoting transcriptional and physiological changes. See text for details.

DOI: https://doi.org/10.7554/eLife.36828.018

*cyp-36A1/nhr-46*-mediated gene expression are parallel pathways downstream of HIF that regulate stress resistance.

## Human cytochrome P450 enzymes might act as mediators of HIF-dependent gene expression

We speculate that some human CYPs might serve as mediators of HIF-dependent gene expression changes through a mechanism analogous to that we describe above for CYP-36A1. In support of this hypothesis, previous studies have identified several human cytochrome P450 enzymes that are putative direct HIF targets based on whole-genome ChIP-chip and ChIP-seq analysis (*Mole et al., 2009*; *Schödel et al., 2011*). We note that the humoral nature of a CYP-generated molecule would make it a candidate mediator of non-autonomous regulation of hypoxia response by the EGLN/HIF pathway, such as is observed in remote ischemic preconditioning (*Cai et al., 2013*; *Olenchock et al., 2016*).

## Cytochrome P450 enzymes might be major players in the hypoxia-response pathway

We previously identified another cytochrome P450 gene, *cyp-13A12*, as acting downstream of *egl-9* in a locomotory behavior (*Ma et al., 2013*). In contrast to CYP-36A1, which is upregulated by HIF-1, CYP-13A12 is downregulated by HIF-1 upon hypoxia exposure or in *egl-9(lf)* mutants. The downstream effectors of CYP-36A1 and CYP-13A12 are also distinct, as we show here that CYP-36A1 regulates a nuclear receptor that controls transcription, whereas CYP-13A12 generates eicosanoids that act on a seconds-to-minutes timescale unlikely to require gene expression changes. Thus, different cytochrome P450 enzymes can act broadly, through multiple mechanisms, downstream of the EGL-9/HIF-1 hypoxia-response pathway in *C. elegans*. We propose that cytochrome P450 enzymes might similarly be important HIF effectors in mammals. Polymorphisms in numerous human cytochrome P450 genes have been associated with cardiovascular disease (*Elbekai and El-Kadi, 2006*; *Rowland and Mangoni, 2014*), for which HIF plays a protective role (*Semenza, 2012*), and with cancers (*Agundez, 2004*), for which HIF contributes to pathogenesis (*Semenza, 2012*). Furthermore, a study of genetic adaptations in humans to the environmentally hypoxic Tibetan plateau identified well-established members of the HIF pathway and, intriguingly, also noted positive selection at two

cytochrome P450 loci (*Simonson et al., 2010*). Together these observations suggest that the cytochrome P450 family of enzymes is important in a wide range of hypoxia-associated contexts in humans. We suggest the presence of a mechanistic link between the canonical HIF pathway and the function of cytochrome P450 enzymes in humans and posit that an understanding of how these highly druggable enzymes (*Schuster and Bernhardt, 2007*) control processes downstream of HIF might reveal new therapeutic avenues for treating a broad array of disorders.

Cytochrome P450 enzymes require oxygen as a substrate and thus might be regulated by oxygen availability. A long-standing issue in the understanding of HIF function is 'range finding,' that is how activity of the EGLN/HIF pathway is modulated such that it responds to different oxygen set points depending on context to drive a diversity of biological outputs (*Ratcliffe, 2013*). We hypothesize that the oxygen sensitivity of HIF-regulated cytochrome P450 enzymes might enable them to function in such a range finding mechanism to narrow the range of oxygen concentrations at which a subset of HIF-regulated physiological changes are activated.

# Materials and methods

### Key resources table

| Reagent type (species) or resource | Designation | Source or reference | Identifiers | Additional information |
|---|---|---|---|---|
| Strain, strain background (*Caenorhabditis elegans*) | AU78 | Dennis Kim | | *agIs9 III* |
| Strain, strain background (*C. elegans*) | CB6088 | Jonathan Hodgkin | | *egl-9(sa307) V hif-1(ia4) V* |
| Strain, strain background (*C. elegans*) | JT307 | Creg Darby | | *egl-9(sa307) V* |
| Strain, strain background (*C. elegans*) | MT20483 | Dengke Ma/Bob Horvitz | | *nIs470 IV* |
| Strain, strain background (*C. elegans*) | MT22836 | this paper | | *cyp-36A1(gk824636) I ; egl-9(sa307) V* |
| Strain, strain background (*C. elegans*) | MT23218 | this paper | | *nIs682 X* |
| Strain, strain background (*C. elegans*) | MT24164 | this paper | | *cyp-36A1(gk824636) I ; agIs9 III* |
| Strain, strain background (*C. elegans*) | MT24165 | this paper | | *cyp-36A1(gk824636) I ; agIs9 III ; egl-9(sa307) V* |
| Strain, strain background (*C. elegans*) | MT24166 | this paper | | *agIs9 III ; egl-9(sa307) V hif-1(ia4) V* |
| Strain, strain background (*C. elegans*) | MT24167 | this paper | | *agIs9 III ; egl-9(sa307) V* |
| Strain, strain background (*C. elegans*) | MT24177 | this paper | | *agIs9 III ; hif-1(ia4) V* |
| Strain, strain background (*C. elegans*) | MT24179 | this paper | | *cyp-36A1(n5666) I ; (nIs470) IV* |
| Strain, strain background (*C. elegans*) | MT24520 | this paper | | *cyp-36A1(n5666) I ; (nIs470) IV ; egl-9(n586ts) V ; nIs674* |
| Strain, strain background (*C. elegans*) | MT24622 | this paper | | *cyp-36A1(n5666) I ; (nIs470) IV ; egl-9(n586ts) V* |
| Strain, strain background (*C. elegans*) | MT24684 | this paper | | *cyp-36A1(gk824636) I ; agIs9 III ; nhr-46(n6126) IV ; egl-9(sa307) V ; nEx2586* |
| Strain, strain background (*C. elegans*) | MT24690 | this paper | | *agIs9 III ; otEx3165* |
| Strain, strain background (*C. elegans*) | MT24692 | this paper | | *agIs9 III ; otEx3156* |

*Continued on next page*

*Continued*

| Reagent type (species) or resource | Designation | Source or reference | Identifiers | Additional information |
|---|---|---|---|---|
| Strain, strain background (C. elegans) | MT24911 | this paper | | cyp-36A1(gk824636) I ; agIs9 III ; nhr-46(n6126) IV ; egl-9(sa307) V |
| Strain, strain background (C. elegans) | MT25047 | this paper | | agIs9 III ; nEx2699 |
| Strain, strain background (C. elegans) | MT25048 | this paper | | cyp-36A1(gk824636) I ; agIs9 III ; nEx2699 |
| Strain, strain background (C. elegans) | MT25050 | this paper | | cyp-36A1(gk824636) I ; agIs9 III ; otEx3156 |
| Strain, strain background (C. elegans) | MT25051 | this paper | | cyp-36A1(gk824636) I ; agIs9 III ; otEx3165 |
| Strain, strain background (C. elegans) | MT25104 | this paper | | cyp-36A1(gk824636) I ; agIs9 III ; nhr-46(n6126) IV ; egl-9(sa307) V ; nEx2713 |
| Strain, strain background (C. elegans) | MT25107 | this paper | | cyp-36A1(gk824636) I ; agIs9 III ; nhr-46(n6126) IV ; egl-9(sa307) V ; nEx2715 |
| Strain, strain background (C. elegans) | MT25196 | this paper | | agIs9 III ; nhr-46(n6125) IV |
| Strain, strain background (C. elegans) | MT25197 | this paper | | cyp-36A1(gk824636) I ; agIs9 III ; nhr-46(n6125) IV ; egl-9(sa307) V |
| Strain, strain background (C. elegans) | MT25211 | this paper | | cyp-36A1(gk824636) I |
| Strain, strain background (C. elegans) | MT25212 | this paper | | cyp-36A1(gk824636) I ; egl-9(n586ts) V |
| Strain, strain background (C. elegans) | MT25213 | this paper | | cyp-36A1(gk824636) I ; nhr-46(n6126) IV ; egl-9(n586ts) V |
| Strain, strain background (C. elegans) | MT25214 | this paper | | nhr-46(n6126) IV |
| Strain, strain background (C. elegans) | MT25215 | this paper | | egl-9(n586ts) V |
| Strain, strain background (C. elegans) | MT25596 | this paper | | cyp-36A1(gk824636) I ; agIs9 III ; egl-9(sa307) V ; nEx2853 |
| Strain, strain background (C. elegans) | MT25599 | this paper | | cyp-36A1(gk824636) I ; agIs9 III ; egl-9(sa307) V ; nEx2856 |
| Strain, strain background (C. elegans) | MT25601 | this paper | | cyp-36A1(gk824636) I ; agIs9 III ; egl-9(sa307) V ; nEx2849 |
| Strain, strain background (C. elegans) | MT25605 | this paper | | cyp-36A1(gk824636) I ; agIs9 III ; egl-9(sa307) V ; nEx2859 |
| Strain, strain background (C. elegans) | MT25606 | this paper | | agIs9 III ; nEx2860 |
| Strain, strain background (C. elegans) | MT25610 | this paper | | cyp-36A1(gk824636) I ; agIs9 III ; nhr-46(n6126) IV ; egl-9(sa307) V ; nEx2864 |
| Strain, strain background (C. elegans) | MT25611 | this paper | | cyp-36A1(gk824636) I ; agIs9 III ; nEx2860 |
| Strain, strain background (C. elegans) | MT25627 | this paper | | agIs9 III ; nhr-46(n6126) IV |
| Strain, strain background (C. elegans) | ZG31 | Huaqi Jiang/Jo Anne Powell-Coffman | | hif-1(ia4) V |

## *C. elegans* strains and transgenes

All *C. elegans* strains were cultured as described previously (*Brenner, 1974*). We used the N2 Bristol strain as the reference wild-type strain, and the polymorphic Hawaiian strain CB4856 (*Davis et al., 2005*) for genetic mapping and SNP analysis. We used the following mutations and transgenes:

LGI: *cyp-36A1(n5666, gk824636)*

LGIII: $agIs219[P_{T24B8.5}::gfp::unc-54\ 3'UTR, P_{ttx-3}::gfp::unc-54\ 3'UTR]$
LGIV: $nhr-46(n6125, n6126), nIs470[P_{cysl-2}::gfp, P_{myo-2}::mCherry], him-8(e1489)$
LGV: $egl-9(n586, sa307), hif-1(ia4)$
LGX: $nIs682[P_{cyp-36A1}::gfp::unc-54\ 3'UTR, P_{myo-3}::mCherry::unc-54\ 3'UTR]$
Unknown linkage: $nIs674[P_{cyp-36A1}::cyp-36A1(+)\ gDNA::cyp-36A1\ 3'UTR, P_{myo-3}::mCherry::unc-54\ 3'UTR]$

## Extrachromosomal arrays

$otEx3156\ [P_{dpy-7}::hif-1(P621A), P_{ttx-3}::rfp], otEx3165\ [P_{unc-120}::hif-1(P621A), P_{ttx-3}::rfp], nEx2699\ [P_{rab-3}::hif-1(P621A)::F2A::mCherry::tbb-2\ 3'UTR, P_{ttx-3}::mCherry], nEx2860\ [P_{vha-6}::hif-1(P621A)::F2A::mCherry::tbb-2\ 3'UTR, rol-6(su1006dm)], nEx2849\ [P_{vha-6}::cyp-36A1\ cDNA::F2A::mCherry::tbb-2\ 3'UTR, rol-6(su1006dm)], nEx2856\ [P_{dpy-7}::cyp-36A1\ cDNA::F2A::mCherry::tbb-2\ 3'UTR, rol-6(su1006dm)], nEx2853\ [P_{rab-3}::cyp-36A1\ cDNA::F2A::mCherry::tbb-2\ 3'UTR, rol-6(su1006dm)], nEx2859\ [P_{unc-54}::cyp-36A1\ cDNA::F2A::mCherry::tbb-2\ 3'UTR, P_{ttx-3}::mCherry::tbb-2\ 3'UTR], nEx2586\ [P_{nhr-46}::nhr-46(+)\ gDNA::nhr-46\ 3'UTR, P_{myo3}::mCherry::unc-54\ 3'UTR], nEx2715[P_{unc-54}::nhr-46\ cDNA::F2A::mCherry::tbb-2\ 3'UTR], nEx2713\ [P_{rab-3}::nhr-46\ cDNA::F2A::mCherry::tbb-2\ 3'UTR], nEx2864\ [P_{vha-6}::nhr-46\ cDNA::F2A::mCherry::tbb-2\ 3'UTR, rol-6(su1006dm)]$

Note on allele usage: For egl-9, the weaker n586 allele was used for the screen and in the initial phenotypic characterization of screen mutants (Figure 1A–1F and Figure 2C–2E). The stronger sa307 allele was used for all other experiments, except for in the slow killing assay, for which the sa307 allele is less protective than weaker alleles, as previously reported (Bellier et al., 2009). For cyp-36A1, the allele identified from the screen, n5666, was used in the initial phenotypic characterization (Figure 1D–1F and 2C–2E); the putative null allele gk824636 was used for all other experiments. Alleles for other genes were used as indicated in the figure legends.

## Molecular biology and transgenic strain construction

The $P_{cyp-36A1}::gfp::unc-54\ 3'UTR$ construct (transgene nIs682) was generated by using PCR fusion (Hobert, 2002) to fuse a PCR product containing the cyp-36A1 promoter fragment (4.4 kb of upstream sequence) to a PCR product containing gfp::unc-54 3'UTR. The cyp-36A1 rescuing construct (transgene nIs674) was generated by amplifying a PCR product from gDNA containing 4.4 kb upstream, the cyp-36A1 locus, and 1.6 kb downstream. The nhr-46 rescuing construct (transgene nEx2586) was generated by amplifying a PCR product from gDNA containing 1.9 kb upstream, the nhr-46 locus, and 0.9 kb downstream. All remaining constructs were generated using the Infusion cloning technique (Clontech). The promoter fragments used for dpy-7 (hypoderm) (Gilleard et al., 1997), vha-6 (intestine) (Allman et al., 2009) rab-3 (neurons) (Mahoney et al., 2006) and unc-54 (muscle) (D. Ma, personal communication) contain 1.3, 0.9, 1.4, and 1.9 kb, respectively, of sequence upstream of the start codons of each of these genes. Expression of tissue-specific rescuing transgenes was confirmed using mCherry tagging. We noticed expression in some unidentified cells outside the body-wall muscle near the junction of the pharynx and intestine in the strain cyp-36A1 $(gk824636);\ agIs9;\ egl-9(sa307);\ nEx2859\ [P_{unc-54}::cyp-36A1\ cDNA::F2A::mCherry::tbb-2\ 3'UTR, P_{ttx-3}::mCherry::tbb-2\ 3'UTR]$; such expression might contribute to rescue of the mutant phenotype in that strain. C45E5.6b was used for nhr-46 cDNA. F38A6.3a with a P621A stabilizing mutation was used for hif-1 cDNA (Pocock and Hobert, 2008). Where present, the F2A sequence served as a ribosomal skip sequence to cause separation of the two peptides encoded before and after the F2A (Ahier and Jarriault, 2014). Transgenic strains were generated by germline transformation as described (Mello et al., 1991). All transgenic constructs were injected at 2.5–50 ng/μl.

## Mutagenesis screen for suppressors of egl-9

To screen for suppressors of the egl-9 egg-laying defect, we mutagenized egl-9(n586) mutants with ethyl methanesulfonate (EMS) as described previously (Brenner, 1974). The starting strain contained the $P_{cysl-2}::gfp\ (nIs470)$ transgene, which is highly expressed in egl-9(lf) mutants and served as a reporter for HIF-1 activity (Ma et al., 2012). We used a dissecting microscope to screen the F2 progeny for suppression of the egg-laying defect (i.e. the Egl phenotype), picking (1) adults that appeared less Egl than egl-9(n586) mutants, and (2) eggs laid by the F2 animals that were at an earlier developmental stage than those laid by egl-9(n586) mutants. Screen isolates were backcrossed

to determine dominant vs. recessive and single-gene inheritance pattern and crossed to *him-8 (e1489); egl-9(sa307) hif-1(ia4)* to test complementation with *hif-1(lf)*. The screen allele *n5666*, which conferred a recessive phenotype and was not allelic to *hif-1*, mapped between SNPs *pkP1052* and *rs3139013* on LGI with SNP mapping (*Davis et al., 2005*) using a strain containing *egl-9(n586)* introgressed into the Hawaiian strain CB4856 (*Ma et al., 2013*). Whole-genome sequencing identified a mutation in *cyp-36A1* in the *n5666* interval, and transgenic rescue demonstrated that this *cyp-36A1* mutation is the causative mutation, as described in the text.

## Mutagenesis screen for suppressors of *cyp-36A1*

To screen for downstream effectors of *cyp-36A1*, we mutagenized *cyp-36A1(gk824636); egl-9 (sa307)* with ethyl methanesulfonate (EMS). The starting strain contained the $P_{T24B8.5}$::*gfp (agIs219)* transgene, which has low expression in *cyp-36A1(gk824636); egl-9(sa307)* mutants and served as a reporter for CYP-36A1 activity. We used a dissecting microscope equipped to examine GFP fluorescence to screen for F2 progeny with high GFP fluorescence and an Egl appearance. The only two isolates failed to complement and were found to be alleles of *nhr-46* by whole-genome sequencing and transgenic rescue. The mutant phenotypes of *cyp-36A1(lf); n6126; egl-9(lf)* were rescued by an *nhr-46(+)* transgene, demonstrating that the mutation in *nhr-46* is the causative mutation and suggesting that *n6126* is a loss-of-function allele.

## Behavioral assays

To quantify egg-laying behavior, we scored the developmental stages of eggs laid by young adult hermaphrodites as described previously (*Ringstad and Horvitz, 2008*). Egg-laying defective mutants retain eggs longer in the uterus, thus laying them at later developmental stages. To examine egg-laying behavior after exposure to hypoxia, young adult animals were placed in a hypoxia chamber (Coy Laboratory) at 1% $O_2$ balanced by $N_2$ for 24 hr, after which the egg-laying assay was performed in normoxia. Control animals were exposed to room air (21% $O_2$) for 24 hr; animals were randomly allocated to 1% or 21% $O_2$ treatment. Locomotion assays were performed on bacterial food and quantified using a custom worm tracker, as described previously (*Paquin et al., 2016*). Defecation assays were performed as described previously (*Thomas, 1990*), counting the number of defecation cycles in ten minutes.

## *Pseudomonas aeruginosa* killing assay

Sensitivity to the *Pseudomonas aeruginosa* strain PA14 was assayed using the big lawn killing assay as described previously (*Reddy et al., 2009*). The big lawn killing assay was used to remove any influence of avoidance behavior on survival, as wild-type PA14 avoidance is dependent on normal aerotaxis behavior (*Reddy et al., 2009*), and *egl-9(lf)* mutants have previously been shown to display abnormal aerotaxis (*Chang and Bargmann, 2008*).

## Tunicamycin survival assay

Sensitivity to tunicamycin was assayed by placing at least 100 starvation-synchronized L1 animals on NGM plates containing 5 μg/ml tunicamycin (Sigma), made using 10 mg/ml tunicamycin stock in DMSO and seeded with *E. coli* OP50 bacteria. Survival to the L4 larval stage or later was determined after three days.

## t-BOOH survival assay

Sensitivity to tert-butyl hydroperoxide (t-BOOH) was assayed by placing ~60 young adult worms on NGM plates containing 7.5 mM t-BOOH, made using 70% t-BOOH solution (Sigma) and seeded with *E. coli* OP50 bacteria. Survival was evaluated after 10 hr.

## Microscopy

Epifluorescence images of $P_{T4B8.5}$::*gfp* expression were obtained using an AxioImager Z2 upright microscope (Zeiss) and ZEN software (Zeiss). Confocal images of $P_{cyp-36A1}$::*gfp* expression were obtained using an LSM 800 instrument (Zeiss) and ZEN software. Fluorescence intensity for the $P_{T24B8.5}$::*gfp* reporter was quantified by measuring average intensity in a 300 μm section of the intestine centered on the vulva using FIJI software. $P_{T24B8.5}$::*gfp* reporter imaging conditions were

optimized for observation of fluorescence in the midbody; fluorescence was not saturated in this region in quantified images.

## RNA isolation for qRT-PCR and RNA-seq

All strains were maintained at 22.5°C for at least two generations without starvation prior to experiment. ~150 very young adults (0–1 eggs in uterus) were picked into M9 buffer and allowed to settle. M9 was aspirated, and worms were then rinsed twice with M9 and twice with RNase-free water. Excess liquid was aspirated and the pellet was frozen in liquid nitrogen. RLT buffer (QIAGEN) was added to the frozen pellet, and worms were lysed using a BeadBug microtube homogenizer (Sigma) and 0.5 mm zirconium beads (Sigma). RNA was extracted using the RNeasy Mini kit (QIAGEN) according to the manufacturer's instructions.

## cyp-36A1 mRNA expression analysis by qRT-PCR

Reverse transcription was performed using SuperScript III (Invitrogen). Quantitative PCR was performed using Applied Biosystems Real-Time PCR Instruments. Expression levels were normalized to the expression of the ribosomal subunit gene rpl-32.

## Primers for qRT-PCR

cyp-36A1 F: ACCAGCTTGTCCAACACCAA
  cyp-36A1 R: CACGCTTTGGCTCCCATTTC
  rpl-32 F: GGCTACACGACGGTATCTGT
  rpl-32 R: CAAGGTCGTCAAGAAGAAGC

## RNA-seq library preparation

RNA integrity and concentration were checked on a Fragment Analyzer (Advanced Analytical). The mRNA was purified by polyA-tail enrichment, fragmented, and reverse transcribed into cDNA (Illumina TruSeq). cDNA samples were then end-repaired and adaptor-ligated using the SPRI-works Fragment Library System I (Beckman Coulter Genomics) and indexed during amplification. Libraries were quantified using the Fragment Analyzer (Advanced Analytical) and qPCR before being loaded for single-end sequencing using the Illumina HiSeq 2000.

## RNA-seq data analysis

Reads were aligned against the *C. elegans* ce10 genome assembly using bwa 0.7.5a (*Li and Durbin, 2009*) and samtools/0.1.19 (*1000 Genome Project Data Processing Subgroup et al., 2009*) (bwa aln/bwa samse), and mapping rates, fraction of multiply-mapping reads, number of unique 20-mers at the 5' end of the reads, insert size distributions and fraction of ribosomal RNAs were calculated using dedicated perl scripts and bedtools v. 2.17.0 (*Quinlan and Hall, 2010*). For expression analysis, reads were aligned against the *C. elegans* ce10 genome/ENSEMBL 65 annotation using RSEM 1.2.15 (*Li and Dewey, 2011*) and bowtie 1.0.1 (*Langmead et al., 2009*), with the following parameters: -p 6 –bowtie-chunkmbs 1024 –output-genome-bam. Raw expected read counts were retrieved and used for differential expression analysis with Bioconductor's edgeR package in the R 3.2.3 statistical environment (*Robinson et al., 2010*). First, common, trended, and gene-specific read dispersion across sequencing libraries and genes was estimated using the estimateDisp function. Given the small number of replicates, a gene-wise negative binomial generalized linear model (GLM) with quasi-likelihood tests (as implemented in the glmQLFit function) was used to test for differential expression between conditions (*Lun et al., 2016*). Briefly, this statistical framework works by first fitting the observed and expected distributions of read counts for each gene across conditions using a GLM, which is based on the negative binomial distribution and the observed read dispersion. The significance of biases in read counts is then tested using the quasi-likelihood F-test (implemented in glmQLFTest). This test provides more robust and reliable error rate control at low number of replicates, because it reflects the uncertainty in read distribution better than the likelihood ratio test. Models were fitted across all conditions and relevant differential expression testing was performed using glmQLFTest between pairs of conditions of interest. P values were adjusted for multiple comparisons using the Benjamini-Hochberg procedure (*Benjamini and Hochberg, 1995*). Gene ontology enrichment analysis was performed using GOrilla (*Eden et al., 2009*), examining genes that were

significantly downregulated in *cyp-36A1(lf); egl-9(lf)* double mutants vs. *egl-9(lf)* single mutants (i.e. orange circle in **Figure 2B**) as compared to genes that were at least twofold upregulated in *egl-9(lf)* mutants vs. wild type (adjusted p value<0.05) and significantly downregulated in *egl-9(lf) hif-1(lf)* vs. *egl-9(lf)* (adjusted p value<0.05) (i.e. purple circle in **Figure 2B**).

### Statistical analysis
Chi-square tests were used to compare the distribution of stages of eggs laid by wild-type and mutant animals. Unpaired t-tests were used to compare *cyp-36A1* mRNA expression between strains, survival on tunicamycin between strains, survival on t-BOOH between strains, and $P_{T24B8.5}::$ *gfp* reporter fluorescence intensity between strains. Log-rank (Mantel-Cox) tests were used to compare survival of different strains on *Pseudomonas aeruginosa*. In cases of multiple comparisons, a Holm-Bonferroni correction was applied. Statistical tests were performed using GraphPad Prism software (Graphpad Prism, RRID:SCR_002798) version 7.0a. Biological replicates were performed using separate populations of animals.

### Accession numbers
The GEO accession number for the RNA-seq dataset in this paper is GSE108283.

## Acknowledgements
We thank K Burkhart, A Doi, S Sando, V Dwivedi, J Saul, E Lee, JN Kong, and D Ghosh for helpful discussion. Some strains were provided by the CGC, which is funded by NIH Office of Research Infrastructure Programs (P40 OD010440).

## Additional information

### Funding

| Funder | Grant reference number | Author |
| --- | --- | --- |
| National Institutes of Health | GM024663 | Corinne L Pender<br>H Robert Horvitz |
| Howard Hughes Medical Institute | | Corinne L Pender<br>H Robert Horvitz |
| National Institutes of Health | T32GM007287 | Corinne L Pender |

The funders had no role in study design, data collection and interpretation, or the decision to submit the work for publication.

### Author contributions
Corinne L Pender, Conceptualization, Formal analysis, Validation, Investigation, Visualization, Methodology, Writing—original draft, Writing—review and editing; H Robert Horvitz, Conceptualization, Supervision, Funding acquisition, Visualization, Methodology, Writing—review and editing

### Author ORCIDs
Corinne L Pender (iD) http://orcid.org/0000-0001-7092-3079
H Robert Horvitz (iD) http://orcid.org/0000-0002-9964-9613

### Decision letter and Author response
Decision letter https://doi.org/10.7554/eLife.36828.025
Author response https://doi.org/10.7554/eLife.36828.026

## Additional files

### Supplementary files

• Supplementary file 1. Genes that were significantly differentially expressed between wild type and *egl-9(lf)* and at least twofold downregulated in *egl-9(lf)* vs. wild type.
DOI: https://doi.org/10.7554/eLife.36828.019

• Supplementary file 2. Genes that were significantly differentially expressed between wild type and *egl-9(lf)* and at least twofold upregulated in *egl-9(lf)* vs. wild type.
DOI: https://doi.org/10.7554/eLife.36828.020

• Transparent reporting form
DOI: https://doi.org/10.7554/eLife.36828.021

### Data availability

Sequencing data have been deposited in GEO under accession code GSE108283.

The following dataset was generated:

| Author(s) | Year | Dataset title | Dataset URL | Database, license, and accessibility information |
|---|---|---|---|---|
| Pender CL, Horvitz HR | 2017 | Hypoxia-inducible factor cell non-autonomously regulates C. elegans stress responses and behavior via a nuclear receptor | www.ncbi.nlm.nih.gov/geo/query/acc.cgi?acc=GSE108283 | Publicly available at the NCBI Gene Expression Omnibus (accession no: GSE108283). |

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
