## [Decision Letter]

Thank you for submitting your article "Hypoxia-inducible factor cell non-autonomously regulates *C. elegans* stress responses and behavior via a nuclear receptor" for consideration by *eLife*. Your article has been reviewed by three peer reviewers, including Matt Kaeberlein as the Reviewing Editor and Reviewer #1, and the evaluation has been overseen by Didier Stainier as the Senior Editor.

The reviewers have discussed the reviews with one another and the Reviewing Editor has drafted this decision to help you prepare a revised submission.

Pender and Horvitz describe a new mechanism by which HIF-1 activity in neurons regulates gene expression in a cell non-autonomous fashion to control egg laying and stress resistance in *C. elegans*. This is a nicely constructed and mostly complete story that makes a compelling case for the model that activation of HIF-1 in neurons directly induces expression of CYP-36A1 in neurons which then acts cell non-autonomously to suppress the activity of NHR-46 in intestine and perhaps other tissues. Suppression of NHR-46 activity by this model then drives both the egg-laying defect and the enhanced resistance to specific stressors including *P. aeruginosa*, tunicamycin, and t-BOOH. All three reviewers appreciated the importance and interest of the study and were generally supportive of publication, with a few significant comments that the authors should be able to address.

Major Comments:

1) What happens in the *nhr-46(lf)* single mutant to egg laying and stress resistance? According to the model, this mutant should show both phenotypes, but I didn't see that data included. If the mutant does not show these phenotypes, how can the current model explain that?

2) The assertions made in the manuscript are generally strongly supported by the data provided, but the evidence that *cyp-36A1* regulates gene expression cell non-autonomously and through the downstream effector *nhr-46* in Figure 3 could perhaps be strengthened. The images shown are intriguing, but the use of only the midbody region, lack of quantification, and the data showing that every transgene rescued the T24B8.5 induction gets very "hand-wavy" to this reviewer. At the very least, it would be nice to have full images, a reproducible method of quantification, and a better explanation for why the midbody would be more important, and that this isn't an artifact of modified transport, delayed signaling, or GFP buildup (each of which could be interesting on their own). The assertion that CYP-36A1 generates a diffusible signal is reasonable, but other explanations exist and the lack of any negative transgenic data is confounding. In addition, there seems (from the images), to be a lot of worm to worm variation, which is unexplained and not statistically or experimentally examined.

3) Related to #2, similar concerns about the lack of quantification and unclear statistical rigor apply to GFP images in other figures. How many times were these experiments replicated? 3-5 animals in one experiment is not sufficient. Were the tissue restrictions on gene expression in tissue-specific lines confirmed in any way?

4) Although the rescue of the *n5666* mutation by a transgene carrying the *cyp-36A1* gene provides strong evidence that the *n5666* mutation is a recessive allele and likely a loss-of-function allele. Given this, does RNAi knockdown of *cyp-36A1* recapitulate the same phenotype? This would seem to be a trivial experiment that would strengthen the authors' model. Is the *n5666* mutation expected to cause a loss of function in the protein, perhaps through disruption of a conserved domain?

---

## [Author Response]

Major Comments:1) What happens in the nhr-46(lf) single mutant to egg laying and stress resistance? According to the model, this mutant should show both phenotypes, but I didn't see that data included. If the mutant does not show these phenotypes, how can the current model explain that?

*nhr-46(lf)* single mutants display wild-type egg laying (added to manuscript; Figure 4D) and nearly wild-type stress resistance (Figures 4E, 4F, and 4G), though they have increased expression of the *P_T24B8.5_::gfp* reporter, similar to *cyp-36A1(lf); nhr-46(lf); egl-9(lf)* mutants (added to manuscript; Figures 4B and 4C). We have updated the manuscript to include this information as well as our interpretation of this interesting result (subsection “A screen for suppressors of *cyp-36A1(lf)* identifies the nuclear receptor gene *nhr-46*”). Specifically, we suggest that at least one additional pathway downstream of *egl-9* acts in parallel to *nhr-46* to regulate *egl-9*-mediated modulation of egg laying and stress resistance, though at least some gene expression changes can be mediated by *nhr-46* alone:

“Interestingly, *nhr-46(lf)* single mutants display wild-type egg laying (Figure 4D), tunicamycin resistance (Figure 4F), and oxidative stress resistance (Figure 4G), and nearly wild-type survival on *Pseudomonas* (Figure 4E and Figure 4—figure supplement 2), indicating that in addition to *nhr-46* at least one other pathway is required to transduce *egl-9*-mediated modulation of egg laying and stress resistance. For example, in *egl-9(lf)* mutants or hypoxia, HIF-1 might drive expression of two (or more) targets that act together to inhibit egg laying and promote stress resistance. Thus, *cyp-36A1* activation and consequent *nhr-46* inhibition would promote egg-laying inhibition and stress resistance in *egl-9(lf)* mutant animals – in which this second pathway was also activated – but not in *nhr-46(lf)* single mutants – in which this second pathway was not activated. *nhr-46(lf)* single mutants have increased expression of the *P_T24B8.5_::gfp* reporter (Figures 4B, 4C and figure 4—figure supplement 1), suggesting that at least some gene expression changes can be mediated by *nhr-46* alone.”

2) The assertions made in the manuscript are generally strongly supported by the data provided, but the evidence that cyp-36A1 regulates gene expression cell non-autonomously and through the downstream effector nhr-46 in Figure 3 could perhaps be strengthened. The images shown are intriguing, but the use of only the midbody region, lack of quantification, and the data showing that every transgene rescued the T24B8.5 induction gets very "hand-wavy" to this reviewer. At the very least, it would be nice to have full images, a reproducible method of quantification, and a better explanation for why the midbody would be more important, and that this isn't an artifact of modified transport, delayed signaling, or GFP buildup (each of which could be interesting on their own). The assertion that CYP-36A1 generates a diffusible signal is reasonable, but other explanations exist and the lack of any negative transgenic data is confounding. In addition, there seems (from the images), to be a lot of worm to worm variation, which is unexplained and not statistically or experimentally examined.

We have updated all images to show whole worms rather than just the midbody. We have also quantified fluorescence for all images and, although there is worm-to-worm variability as the reviewers note, we find statistical significance for all previously-described differences in fluorescence intensity.

Regarding the importance of midbody expression, as can now be observed with the whole-worm images, we do see differences in the expression in the foregut and hindgut as well. However, expression of the reporter is higher in the foregut and hindgut than in the midgut in all strains, including the wild type (as shown previously, Shivers et al., 2010), such that expression changes are most easily observed in the midbody. To quantify fluorescence, we assessed average intensity in a 300 μm section of the intestine centered on the vulva, as is now described in Materials and methods.

Regarding the lack of negative transgenic data, we hypothesize that expressing *cyp-36A1* in a much more restricted set of cells would not be sufficient to rescue *cyp-36A1(lf)*, more because of a lower overall level of expression than because of a specific localization. We do not believe that such experiments would add substantially to this manuscript. The expression patterns for rescue constructs were confirmed as described below.

3) Related to #2, similar concerns about the lack of quantification and unclear statistical rigor apply to GFP images in other figures. How many times were these experiments replicated? 3-5 animals in one experiment is not sufficient. Were the tissue restrictions on gene expression in tissue-specific lines confirmed in any way?

We agree with these concerns. We have increased the number of animals per experiment to 10, have performed each experiment in duplicate, and have quantified and performed statistical analysis on fluorescence intensity, as described above and in Materials and methods.

Hypoderm and muscle-specific HIF-1-expressing strains have been characterized previously (Pocock and Hobert, 2008). For all other tissue-specific constructs, we have added an mCherry tag to the relevant cDNA and confirmed correct expression, as described in the Materials and methods.

4) Although the rescue of the n5666 mutation by a transgene carrying the cyp-36A1 gene provides strong evidence that the n5666 mutation is a recessive allele and likely a loss-of-function allele. Given this, does RNAi knockdown of cyp-36A1 recapitulate the same phenotype? This would seem to be a trivial experiment that would strengthen the authors' model. Is the n5666 mutation expected to cause a loss of function in the protein, perhaps through disruption of a conserved domain?

The *gk824636* nonsense allele of *cyp-36A1* causes the same mutant phenotype as *n5666* (Figure 3—figure supplement 3C). The residue altered by the *n5666* mutation is indeed a conserved glycine (e.g. conserved in the closest human homolog CYP2A6), suggesting that it is important for CYP-36A1 function. These observations strongly support our statement that *n5666* is likely a reduction-of-function allele. RNAi knockdown of *cyp-36A1* does not recapitulate the *n5666* mutant phenotype; however, RNAi-mediated knockdown might not sufficiently reduce CYP-36A1 activity to cause the same phenotype as the mutant.